# The Starch Physicochemical Properties between Superior and Inferior Grains of Japonica Rice under Panicle Nitrogen Fertilizer Determine the Difference in Eating Quality

**DOI:** 10.3390/foods11162489

**Published:** 2022-08-17

**Authors:** Yan Jiang, Yue Chen, Can Zhao, Guangming Liu, Yi Shi, Lingtian Zhao, Yuan Wang, Weiling Wang, Ke Xu, Guohui Li, Qigen Dai, Zhongyang Huo

**Affiliations:** Jiangsu Key Laboratory of Crop Genetics and Physiology, Jiangsu Key Laboratory of Crop Cultivation and Physiology, Jiangsu Co-Innovation Center for Modern Production Technology of Grain Crops, Agricultural College, Yangzhou University, Yangzhou 225009, China

**Keywords:** panicle nitrogen fertilizer, japonica rice, superior and inferior grains, physicochemical properties of starch, starch fine structure

## Abstract

Nitrogen fertilizer is essential for rice growth and development, and topdressing nitrogen fertilizer at panicle stage has a huge impact on rice grain quality. However, the effect of panicle nitrogen fertilizer (PNF) on starch physicochemical properties and fine structure remain unclear. In this study, four PNF levels (0, 60, 120, 180 kg N ha^−1^) were grown with the same basal and tiller fertilizer (150 kg N ha^−1^). The starch physicochemical properties, fine structure, texture properties and eating quality of two japonica rice were determined. We found that the content of total protein, crude fat and amylose between superior and inferior grains were significantly different. Compared with inferior grains, superior grains had low relative crystallinity, good pasting characteristics and outstanding eating quality. With the increase of nitrogen application rates, the starch volume mean diameter was lower; the average chain length of amylopectin was longer; and the relative crystallinity of starch was higher. The changes above in starch structure resulted in an increase in starch solubility, swelling power and gelatinization enthalpy, and led to a decrease in retrogradation enthalpy, retrogradation percentage and pasting viscosity, consequently contributing to the increase in hardness and stickiness of rice and the deterioration of taste value. These results indicated that topdressing PNF lengthened the amylopectin chain, decreased starch granule size, enhanced crystallization stability and increased gelatinization enthalpy, which were the direct reasons for the deterioration of cooking and eating quality.

## 1. Introduction

Rice (*Oryza sativa* L.) is a widely cultivated cereal crop in Asia, and about half of the world’s population depends on rice and its related products for energy and nutritional needs [1]. In order to balance food supply and population growth, China has greatly increased rice yield in the past few decades through policy guidance, variety renewal, and cultivation regulation [2]. However, with increasing living standards, the demand for rice with good taste is rapidly increasing. Starch represents about 90% of the dry weight of mature head rice, suggesting that starch plays an important role in determining eating quality [3]. The formation of starch structure is complicated. First, α-D-glucose is connected by α-1,4 glycosidic chain to form a single starch chain, then the branched chain is connected by α-1,6 glycosidic chain to form amylose and amylopectin molecules under enzymatic reaction, and adjacent amylopectin branches form crystalline clusters, which then stack alternately to form semi-crystalline growth rings, and eventually develop into starch granules [4,5]. Starch fine structure refers to the molecular structure of starch chemical components such as amylose, amylopectin and lipids and the complex structure formed by the interaction between these components, covering almost all the characteristics of starch [6]. 

The fine structure of starch is related to the physicochemical properties and quality of rice. Deng et al. [7] found that higher molecular weight, mean diameter and relative crystallinity of ratoon rice were regarded as the main reasons for its high gelatinization enthalpy, retrogradation enthalpy and swelling power compared to first harvest rice. Li et al. [8] found that starch with a higher content of branch chains with a degree of polymerization (DP) 10–26 in amylopectin had a more regular arrangement of crystal lamellae, which resulted in slower starch digestibility. Zhou et al. [9] reported that the proportion of B3 chains (DP > 37) in amylopectin was significantly negatively correlated with starch breakdown value, and significantly positively correlated with pasting temperature, gelatinization enthalpy and setback value. Hardness is a vital indicator for evaluating the texture of rice. Li and Gilbert [10] found that amylose molecules might entangle or co-crystallize with amylopectin branches in the crystalline lamellae, thereby limiting starch swelling and increasing the hardness. 

The palatability of rice is determined by a combination of attributes, including color, smell, taste, stickiness, and hardness [11]. It is generally believed that rice with low amylose content (8–12%) is soft, sticky, glossy, and does not harden rapidly after cooling, which is more in line with consumers’ eating habits in China [12]. Therefore, related indicators such as amylose content and gel consistency were often used to evaluate the palatability of rice. However, in addition to amylose, the physicochemical properties of starch were also related to the palatability of rice, including swelling properties, gel properties and pasting properties of starch molecules when heated in water, as well as retrogradation properties after cooling [13,14]. The good gelatinization properties of starch in rice cooking is one of the indicators of palatability. Higher peak viscosity, breakdown value and gelatinization enthalpy of starch during gelatinization is associated with hybrid rice with a softer and stickier texture [15]. Starch fine structure, including granule size distribution and chain length distribution, plays an important role in the palatability of rice. Starch with thin lamellae and small granule size could form a loose starch gel network, make starch gelatinize with high viscosity and become hard to regenerate after cooling, contributing to the palatability of rice [16]. Rice varieties with more long chains and less short chains in amylopectin have a harder rice texture. Li et al. [17] found that long amylopectin chains and amylose chains may entangle, which slowed down the swelling process of starch, finally making rice more elastic and less sticky.

The eating quality of grains differed greatly depending on the position of grain in the panicle [18,19]. In general, the spikelets that flowered early on the primary stem at the top of the panicle were defined as superior grains, while the spikelets that flowered late on the secondary stem at the base were defined as inferior grains [20]. Rice superior grains had lower amylose content, and starch exhibited higher breakdown values and lower setback values during gelatinization [21]. Ma et al. [22] showed that amylopectin of superior grain had a high proportion of long chains (DP 13–36), and the cooked rice had good resilience and high balance, which meant overall better palatability. Zhu et al. [23] chose rice varieties with different amylose content as materials, and found that the amylopectin of superior grains had more long chains and less short chains, and had more slow-digestible and resistant starches than inferior grains. These findings indicated that superior and inferior grains had significant differences, and rational utilization of these differences could improve grain varieties and starch functionality. At present, there are many studies on the effect of amylopectin chain length distribution characteristics on the eating quality of rice, while the differences in the amylopectin chain length distribution characteristics and starch physicochemical properties of superior and inferior grains remain to be further clarified.

Nitrogen is of paramount importance in determining the growth and development of rice, and rice absorbs nitrogen from soil through the root system. Increasing nitrogen application is the main approach to meet the nitrogen demand of rice. Appropriate nitrogen application should take both yield and rice quality into account, as excessive nitrogen application leads to a decrease in the eating quality [24]. The panicle initiation stage is the starting point of rice reproductive growth, marking the beginning of the differentiation of panicle. The spikelet differentiation stage is the key period for the formation of spikelet and panicle structure, which is related to the number of spikelets in the rice population. Therefore, in production, panicle fertilizer is usually top-dressed at the panicle initiation stage and spikelet differentiation stage to ensure a suitable population of spikelets and to promote grain filling. Japonica rice is favored by consumers and is widely planted in China. It has characteristic short, rounded grains, lower protein and amylose content, and generally better processing and eating quality compared to indica rice [25].

To date, there have been relatively few studies on the effect of PNF on the quality and starch characteristics of japonica rice. Lu et al. [26] reported that topdressing PNF increased the protein content in grains of japonica rice, and reduced the amylose content, gel consistency and pasting viscosity, thus negatively affecting the taste of the rice. The gel consistency reflects the mobility and ductility of the cooled rice flour gel, which related to rice eating quality [27]. Yang et al. [28] applied nitrogen fertilizer during panicle initiation stage, and found that the mean size of starch granules of japonica rice and the proportion of amorphous region decreased, while the relative crystallinity and double helix content increased. However, the differences between the superior and inferior grains of japonica rice under different PNF treatments, and the effects of different PNF on the starch physicochemical properties and structure of the superior and inferior grains of japonica rice have not been examined in depth. 

In this study, Nanjing 9108 and Nanjing 0212 were chosen as experimental cultivars for their different sensory properties. The differences in starch physicochemical properties, fine structure and eating quality of superior and inferior grains of japonica rice, as well as the regulatory changes in nitrogen panicle fertilizer were investigated. The reasons for the differences were discussed in order to provide new ideas for cultivation measures to optimize rice eating quality and starch functional properties.

## 2. Materials and Methods

### 2.1. Experimental Site and Design

The experiment was carried out in 2021 at the experimental base of the Agricultural College of Yangzhou University (32°31′ N, 119°55′ E). The soil type was waterlogged paddy soil with a sticky texture. The pH value of the 0–20 cm soil layer was 7.79, and contained 31.72 g·kg^−1^ of organic matter, 104.91 mg·kg^−1^ of alkaline hydrolysis nitrogen, 119.08 mg·kg^−1^ of available potassium, and 15.40 mg·kg^−1^ of available phosphorus. Nanjing 9108 and Nanjing 0212 were chosen as materials, which are widely cultivated in Jiangsu province and had significant differences in eating quality. Nanjing 9108 has low grain amylose content and good sensory properties, while Nanjing 0212 has a medium eating quality. They had similar heading date and growth period when planted in Yangzhou. Seeds were sown on May 21, and the seedlings with consistent growth were transplanted on 13 June with a row spacing of 30 cm × 12 cm, and 4 seedlings per hole.

The experiment was arranged in a split-plot design, with the amount of PNF as main plot factor. Four levels of PNF were set, namely 0 N (0 kg N ha^−1^), 60 N (60 kg N ha^−1^), 120 N (120 kg N ha^−1^) and 180 N (180 kg N ha^−1^). The source of nitrogen was urea. Half of the nitrogen was applied at the panicle initiation stage, and the other half was applied at the spikelet differentiation stage. Through the investigation of growth process, Nanjing 9108 was applied PNF on 22 July and 3 August, and Nanjing 0212 was one day later than Nanjing 9108, respectively. Variety was the within-plot factor, and the area was 4 m × 4 m. An application of 90 kg N ha^−1^ was conducted before transplanting, and an application of 60 kg N ha^−1^ was conducted at the tiller stage. Each treatment was performed in triplicate, and the inter-plot ridges were wrapped with plastic membrane to ensure separate irrigation and drainage. 100 kg ha^−1^ P_2_O_5_ (as calcium superphosphate, P_2_O_5_ content 13.5%) and 150 kg ha^−1^ K_2_O (as potassium chloride, K_2_O content 52.4%) were applied once as basal fertilizers before transplanting. Cultivation measures such as water management and control of pests and weeds were implemented in accordance with corresponding high-yield cultivation requirements.

### 2.2. Sampling

At the heading stage, 300 spikes that were flowering on the same day were chosen and tagged. The flowering date of each spikelet was recorded and marked at the flowering stage. The superior grains were the spikelets that bloomed on the first and second days, which were mainly concentrated on the primary stems at the top of the spike; the inferior grains were the spikelets that bloomed on the last two days, mainly concentrated on the secondary stems at the base of the spike. At maturity, 60 g each of superior grains and inferior grains were detached from the marked spikelets, dried at 35 °C to 14% moisture, and then stored at 4 °C for measurement.

### 2.3. Measurements and Methods

#### 2.3.1. Determination of Starch, Protein and Crude Fat Content

The total starch content was determined by the total starch kit (K-TSTA, Megazyme, Wicklow, Ireland), the amylose content was measured by spectrophotometer with reference to the national standards of the People’s Republic of China (GB/T 17891-2017), and the amylopectin content was from the total starch content subtracted by the amylose content. According to the method of Tao et al. [29], the total protein content was determined by means of the Kjeldahl method with an automatic Kjeldahl nitrogen analyzer (Kjeltec 8200, Foss, Sweden). The protein components extraction referred to the method of Liu et al. [30], in which albumin, globulin, prolamin and glutenin were continuously extracted and separated, using Coomassie brilliant blue G-250 colorimetry to determine the content of the first three proteins, and using Biuret colorimetry to determine the content of glutenin. Crude fat content was examined by Soxhlet extractor (SOX406, Hanon, China) with reference to the national standards of the People’s Republic of China (GB 5009.6-2016). Content of each component was converted into dry basis content according to moisture content.

#### 2.3.2. Starch Isolation

Brown rice was frozen and then ground with a mixed-type mill (MM 400, Retsch, Germany), then 10 g brown rice flour was weighed and placed in a 50 mL conical flask with 30 mL of 10^−5^ mol L^−1^ sodium hydroxide solution, 0.10 g alkaline protease (Solarbio, China) and two glass balls. The conical flask was well-mixed to disperse the rice flour in the NaOH and protease solution, sealed and then put in a shaker for 24 h at 42 °C to remove protein. The mixture of starch and protease was sieved through a 200-mesh sieve and transferred to a 50 mL centrifuge tube. Deionized water (30 mL) was added, and the centrifuge tube was shaken at 3000 rpm for 10 min. The supernatant was discarded, and the upper layer, yellowish residue mainly composed of unreacted protease, was scraped off. The operation was repeated 3–5 times until the upper layer was free of residue. The sediment was then washed with a mixture of ethanol and chloroform/methanol (1:1 *v*/*v*) to remove fat. The starch was dried in an oven at 60 °C for 24 h, and was stirred and dispersed by a glass rod and then be dried for another 24 h. Finally the starch was sieved through a 200-mesh sieve to obtain native starch and stored in a drying container.

#### 2.3.3. Analysis of Starch Granule Size Distribution

The analysis of starch granule size distribution was conducted by a laser particle size analyzer (MS-2000, Malvern, UK). The starch samples were dispersed in absolute ethanol, and the rotational speed was set to 2500 rpm. The samples were ultrasonically dispersed for 30 s before measurement. The measuring range of the instrument is 0.1–2000 μm. Once the background interference had been eliminated, the diameter of starch samples was determined based on the volume distribution.

#### 2.3.4. Analysis of Amylopectin Chain Length Distribution

According to the method described by Wu et al. [31], starch was debranched by 8-aminopyrene-1,3,6-trisulfonate (APTS). Separations were performed in N-CHO-coated capillaries with carbohydrate separation buffer at 25 °C and 30 kV. Measurements were performed using a gel capillary electrophoresis apparatus (PA-800 Plus, Beckman-Coulter, Brea, CA, USA) equipped with a solid-state laser-induced fluorescence (LIF) detector and an argon ion excitation source.

#### 2.3.5. Analysis of Starch X-ray Diffraction (XRD)

The XRD analysis of starch and the calculation of relative crystallinity were based on the method of Zhang et al. [32], and the X-ray diffraction pattern was obtained using a polycrystalline X-ray diffraction analyzer (D8 Advance, Bruker-AXS, Germany). The measurement voltage was 40 kV, the current was 200 mA, and the diffraction angle of scanning region (2θ) ranged from 4° to 40° at a scanning rate of 0.02°. The X-ray diffraction patterns were processed with MDI Jade 6.0 to obtain the area of crystalline region (A_c_) and the area of amorphous region (A_a_). The relative crystallinity was determined using the following equation: (1)$$\mathrm{Relative}\;\mathrm{crystallinity}=\frac{A_{c}}{A_{c}+A_{a}}\times 100\%$$

#### 2.3.6. Analysis of Starch ^13^C CP/MAS NMR 

According to the method described by Tan et al. [33], starch was analyzed on a solid state nuclear magnetic resonance (Advance III 400WB, Bruker, Germany) at B_0_ = 9.4T. Starch samples were loaded into a 4 mm diameter PSZ (partially stabilized zirconium oxide) rotor with KelF end caps, rotated at 5 kHz with an acquisition time of 50 ms and a minimum of 2400 cumulative scans per spectrum. The preparation of amorphous starch followed the method of Tan et al. [33]. The spectra was peak fitted using software Peakfit (Version 4.12, Palo Alto, CA, USA), and the C1 region (95–105 ppm) was selected to calculate the amorphous content, single helix and double helix content.

#### 2.3.7. Analysis of Starch Solubility and Swelling Power

The determination of starch solubility and swelling power was conducted as described by Cao et al. [34] with some modifications. A 30 mg starch sample (m_0_) was weighed and placed in a 2 mL centrifuge tube (m_1_) with 1 mL of ultrapure water added, and then bathed at 90 °C for 1 h with shaking. The sample was cooled to room temperature (25 °C), then centrifuged at 5000 rpm for 10 min. The supernatant was removed by pipette and the total weight of the residue and centrifuge tube (m_2_) was gauged. Finally, the sample was dried in an oven at 60 °C to constant weight (m_3_). According to Cao et al. [34], the solubility and swelling power were calculated using Equations (2) and (3).
(2)$$\mathrm{Solubility}\;\left(\%\right)=\frac{100\times \left(m_{0}+m_{1}-m_{3}\right)}{m_{0}}$$
(3)$$\mathrm{Swelling}\;\mathrm{power}\;\left(g/g\right)=\frac{m_{2}-m_{1}}{m_{3}-m_{1}}$$

#### 2.3.8. Determination of Starch Thermal Properties

The analysis of starch thermal properties was measured by differential scanning calorimeter (DSC 214, Netzsch, Germany) according to the method described by Hu et al. [35]. A 5 mg starch sample was weighed and placed in an aluminum pan (Netzsch, D-95100, Bayern, Germany). The samples were shaken with 10 μL ultrapure water and sealed, then held at 4 °C for 12 h. Before analysis, the sealed pans were allowed to equilibrate to room temperature before the thermal properties were determined. The instrument was calibrated with the standard vacuum aluminum pan blank. The heating rate was 10 °C min^−1^. The starch was heated from 20 °C to 100 °C to determine the gelatinization characteristics and from 10 °C to 90 °C to determine retrogradation characteristics. Retrogradation percentage (R) was determined using the following equation:(4)$$R\;\left(\%\right)=\mathrm{gelatinization}\;\mathrm{enthalpy}/\mathrm{retrogradation}\;\mathrm{enthalpy}\;\times 100\%$$

#### 2.3.9. Determination of Starch Pasting Properties

The pasting properties were determined using a rapid viscosity analyzer (RVA-3D, Newport Scientific, Australia). A 3 g sample of milled rice flour with 12% moisture content was weighed and placed in an aluminum RVA can and 25 g of ultrapure water was added. The stirring speed was 960 rpm in the initial 10 s, and then kept at 160 r min^−1^. The temperature in the tank started at 50 °C for 1 min and then increased from 50 °C to 95 °C at 12 °C min^−1^, held at 95 °C for 2.5 min, decreased to 50 °C and held 2 min. The peak viscosity, trough viscosity, final viscosity, breakdown value (peak viscosity–trough viscosity), setback value (final viscosity–peak viscosity) and pasting temperature were analyzed by TCW (Thermal Cline for Windows) program.

#### 2.3.10. Analysis of Taste Value and Texture Properties

The taste value was determined by the Satake taste meter (RCTA 11A, Satake Co., Hiroshima, Japan). The rice was hulled and milled, and the broken rice was removed to obtain head rice. Using the method of Zhou et al. [9], 25 g head rice was weighed and washed three times with 1.3 times of water, and then soaked, steamed, kept warm, air-dried, and left to stand [16]. An amount of 8 g cooked rice was weighed, and pressed into an iron ring to determine the taste value. 

The texture properties were measured with a physical property analyzer (TA.XT.PlusC, Stable Micro Systems, UK) with the P/36R probe. The cooked rice grains were placed in the center of the sample table, and the program matched with the force sensing element ran to determine the firmness and stickiness properties.

### 2.4. Statistical Processing and Analysis

All variable data were the average of three experimental values and are shown in the table as mean ± standard deviation. SPSS Statistics 16.0 was used to conduct analysis of variance. In the analysis, the sources of variation included nitrogen (N), variety (V), position (P), and the interaction (N × V, N × P, P × V, and N × V × P). Duncan’s new multiple range method was used to conduct multiple comparisons of variables with statistically significant differences. The data of the same variety in the same column marked with different lowercase letters meant significant difference at *p* < 0.05.

## 3. Results and Discussion

### 3.1. Content of Starch, Protein and Crude Fat

Starch was the main content in the endosperm of rice, followed by protein and crude fat. According to Appendix A, nitrogen, variety, and position had significant effects on the content of starch, and some indicators had the two-factor interactions and three-factor interactions significant effect. The amylose content of superior grains was higher than that of inferior grains, the total starch content and amylopectin content were lower than those of inferior grains, and the ratio of amylose/amylopectin was significantly higher than that of inferior grains (Table 1). The amylose content and the ratio of amylose/amylopectin of superior grains of both varieties were 4.82–29.68% and 9.19–38.70% higher than those of inferior grains, while the total starch and amylopectin contents were 0.86–3.21% and 2.74–9.86% lower, respectively. 

The grain filling of superior grains started early with high activity of granule-bound starch synthase (GBSS), and the inferior grains maintained relatively high activities of ADP glucose pyro-phosphorylase (AGPase) and starch branching enzyme (SBE) in the middle and late stages of grain filling, which contributed to the differences in starch accumulation between superior and inferior grains [36]. 

Topdressing with PNF decreased the content of total starch, amylose, amylopectin and the ratio of amylose/amylopectin, which was consistent with previous studies [24]. Compared with 0 N treatment, the total starch, amylose, and amylopectin content, and the ratio of amylose/amylopectin were significantly lower under high nitrogen (120 N and 180 N) treatments, which were 4.29–8.78%, 7.39–17.58%, 3.70–7.77% and 3.84–10.59%, respectively (Table 1). The phenomenon could relate to the starch synthesis-related enzymes and the expression of related genes. A previous study revealed that the activities of soluble starch synthase (SSS) and GBSS were lowered by nitrogen fertilizer in the later stage, and the mRNA expression levels of OsGBSSI and OsSBEII genes were down-regulated, thereby inhibiting starch accumulation [37]. Xiong et al. [38] found that increasing nitrogen fertilizer application at the tillering stage also decreased amylose content, and the arrangements of amyloplasts in rice were more compact, with fewer interspaces between each other. However, the effect of topdressing at the tillering stage was not as obvious as that at the panicle initiation stage.

It can be seen from Appendix A that nitrogen, variety, and position had a significant effect on the content of protein and crude fat, except for variety in the content of globulin. The total protein content, albumin, globulin, prolamin, glutenin and crude fat of superior grains were lower than those of inferior grains (Table 2). The differences in total protein content, albumin and crude fat reached the significant level, while there was no significant difference in prolamin and glutenin in some treatments. Crude fat content showed the greatest difference, with inferior grains having 46.55–71.38% higher fat content than superior grains. Topdressing with PNF significantly decreased the crude fat content. In addition, the crude fat content of high nitrogen (120 N and 180 N) treatments decreased by 33.74–35.67% compared with 0 N treatment, but when compared with 60 N treatment, crude fat content of high nitrogen treatments only decreased by 0.84–15.01%, which suggested that the crude fat had an insensitive response to the gradient of nitrogen fertilization, which was probably due to the dominating distribution of fat in the germ and scarce distribution in the endosperm.

The content of total protein, albumin and gluten of superior grains were 5.93–21.62%, 9.77–21.38% and 3.90–15.24% lower than those of inferior grains, respectively. Topdressing PNF significantly increased the total protein content, and the content of protein component showed an increasing trend, which was significantly higher in 120 N and 180 N treatments compared with 0 N treatment. Starch and protein competed for carbon source for synthesis. Topdressing with PNF promoted nitrogen accumulation in plants, activated protein synthesis-related enzyme, and accelerated protein synthesis [38]. Under the high nitrogen (120 N and 180 N) treatments, the total protein content increased by 19.65–31.21% compared with 0 N treatment, and the prolamin and glutenin increased by 13.82–24.72% and 18.71–34.92%, respectively, which indicated that the increase of total protein content by PNF was mainly related to prolamin and glutenin. 

### 3.2. Starch Granule Size Distribution

According to Appendix A, nitrogen, variety, position, two-factor interactions, and three-factor interactions had significant effect on the starch granule size distribution. The starch granule size distribution of superior and inferior grains of japonica rice appeared as a bimodal distribution between 0–2 μm and 2–16 μm, which was significantly different under different PNF (Figure 1). Starch granule size was classified into small starch granules (≤2 μm), medium starch granules (2–5 μm) and large starch granules (≥5 μm) according to Hu et al. [39]. The starch granule size distribution of superior and inferior grains of both varieties differed. 

In Nanjing 9108, superior grains had a higher level of small and medium starch granules (≤5 μm) than inferior grains, and the content of large starch granules and starch volume mean diameter were less than those of inferior grains (Table 3). However, the superior grains of Nanjing 0212 showed the opposite trend, which was probably due to cultivar characteristics and enzyme activities related to starch synthesis in superior and inferior grains [23]. 

Topdressing with PNF resulted in an increase in the proportion of small and medium starch granules, and a decrease in the proportion of large starch granules and a decrease in the starch volume mean diameter. Compared with 0 N treatment, the small and medium starch granules of high nitrogen (120 N and 180 N) treatments increased by 4.08–12.88% and 6.31–11.39%, while the content of large starch granules and the starch volume mean diameter decreased by 5.72–9.35% and 3.94–4.97%, respectively, all of whose differences reached the significant level. Later nitrogen application postponed the grain filling. Large starch granules mostly appeared in the early filling stage, and decomposed into small and medium starch granules at the later stage of grain filling, which contributed to the decline in the content of large starch granules and volume mean diameter after topdressing PNF [27].

### 3.3. Amylopectin Chain Length Distribution

From the analysis of variance, nitrogen and variety had significant effect on A chain, B1 and B2 chain. Position has no significant effect on the B1 chain, B3 chain and average DP (Appendix A). The chain lengths of amylopectin in superior and inferior grains of japonica rice under different PNF treatments showed a bimodal distribution, and each treatment reached its peak when the DP was 12 (Figure 2). The chain length distribution of amylopectin was classified into A chain (DP 6–12), B1 chain (DP 13–24), B2 chain (DP 25–36) and B3 chain (DP > 36) according to previous studies [40], where the A and B1 chains were short chains, and the B2 and B3 chains were long chains. The chain length distribution of amylopectin between superior and inferior grains of japonica rice was basically identical for the two varieties studied. Compared with inferior grains, the proportion of B3 chain of superior grains was 1.10–2.48% higher, (A + B1)/(B2 + B3) was 1.01–3.17% lower, and the average chain length was 0.66–1.25% higher under high nitrogen (120 N and 180 N) treatments, which showed few differences between varieties (Table 4). The similarity of chain length distribution and the response to nitrogen application rate for both varieties indicates that genetic background was the main factor regulating the distribution of amylopectin chain length under similar cultivation measures, which was consistent with the conclusion of Zhu et al. [23] that the distribution characteristics of amylopectin chain length between superior and inferior grains in rice were primarily associated with cultivar characteristics. 

Topdressing PNF decreased the content of A chain and increased the content of B2 chain and B3 chain, increasing the average chain length of amylopectin and reducing (A + B1)/(B2 + B3). Compared with 0 N treatment, the A chain content and (A + B1)/(B2 + B3) of amylopectin in high nitrogen (120 N and 180 N) treatments decreased by 3.54–6.72% and 5.45–9.65%, respectively, while the B3 chain content and average chain length increased by 6.10–12.22% and 1.86–4.53%, respectively, all of which showed significant difference. 

The same conclusion was drawn by Yang et al. [41], and the effect was regarded to be associated with the inhibition of SBE enzyme activity and the promotion of SSS enzyme activity by nitrogen through fitting Wu-Gilbert’s chain length distribution model. McMaugh et al. [42] reduced the expression of SSI protein by RNAi technology, and the results indicated that the proportion of extremely short chains (DP 6–7) and intermediate chains (DP 13–22) of amylopectin increased, while the proportion of short chains (DP 8–12) decreased. In addition, previous studies reported that SSIIa in rice endosperm SSS was mainly responsible for elongating amylopectin A chain, while SSIIIa accounted for elongating B2 chain and B3 chain. The difference in chain length distribution was presumably related to the activities of these enzymes and the expression levels of related genes [43,44]. 

Tang et al. [45] isolated granules of different sizes from barley starch, finding the average chain length of amylopectin decreased as the starch granule size decreased, which indicated that the influence of PNF on the chain length distribution of amylopectin could be associated with the differences in the chain length distribution of amylopectin in rice starch granules with different sizes, and further research was necessary.

### 3.4. Starch Crystalline Structure

#### 3.4.1. XRD Pattern and Relative Crystallinity

The X-ray diffraction pattern of starch from superior and inferior grains of japonica rice under different nitrogen fertilizers showed two strong peaks at 15° and 23°, and unresolved doublet peaks at 17° and 18° (2θ), which were typical A-type starch crystals. It could be concluded that PNF did not change the starch crystal type (Figure 3A,B). According to Appendix A, the relative crystallinity was significant affected by nitrogen, variety, position, and N × V. Further analysis of the relative crystallinity (Table 5) suggested that the relative crystallinity of superior grains was lower than that of inferior grains, and the superior grains were 2.89–4.17% significantly lower than the inferior grains under the treatments of topdressing PNF. This was related to the higher amylose content in superior grains. Amylose was observed in starch crystalline lamellae, which meant that amylose molecules could disrupt the double helix structure, thereby reducing relative crystallinity [46]. 

The relative crystallinity of starch from the superior and inferior grains of japonica rice increased as nitrogen application rates increased. Compared with 0 N treatment, the relative crystallinity of inferior grains under treatments of topdressing PNF increased by 1.87–6.08%, and that of superior grains under 180 N increased by 2.47–4.73%, all of which were significantly different. However, Zhu et al. [47] found that increasing the nitrogen fertilizer input before transplanting and tillering stage, under the same total nitrogen application, significantly reduced the relative crystallinity of starch. This indicated that the crystallinity of starch might be related to late-stage nitrogen fertilizer. In this study, the relative crystallinity increased with the decrease of amylose content, implying that relative crystallinity was negatively correlated with amylose content. In addition, the distribution of amylopectin chain length was also strongly associated with the relative crystallinity. More amylopectin A chains would be expected to reduce starch crystallinity [48]. In this study, A chains of amylopectin from superior and inferior grains decreased significantly after topdressing PNF, which was also responsible for the increase of relative crystallinity.

#### 3.4.2. ^13^C CP/MAS NMR Analysis

The irregularly arranged or short-range helical structures could not be detected by XRD, but detected by ^13^C solid-state NMR [49]. Therefore, in order to further clarify the differences in starch crystal structure, the helical structure of starch was analyzed by ^13^C solid-state NMR technique. The native starch spectra was separated into an ordered sub-spectra at the 84 ppm position (Figure 3C,D) as described by Tan et al. [33]. Nuclear magnetic resonance imaging showed that there were four signal regions in starch of japonica rice, namely C1, C4, C2, 3, 5 and C6. The triple peaks in C1 region (94–105 ppm) indicated that the crystal structure was type A. The C1 region had no overlapping signals for other carbon sites, so the region was selected to calculate starch amorphous region, single helix and double helix content. Nitrogen, variety, position, and V × P had significant effect on amorphous, single helix, and double helix (Appendix A). The amorphous region of superior grains was higher than that of inferior grains, while the content of single helix and double helix was lower than that of inferior grains (Table 5). The amorphous region of superior grains under high nitrogen (120 N and 180 N) treatments was 2.29–7.07% higher than that of inferior grains, while the single helix and double helix contents were 2.34–6.53% and 12.15–22.99% lower, respectively, all of which were significantly different. 

Topdressing PNF decreased the starch amorphous region and single helix content, increased the double helix content, and the effect was gradually obvious with the increase of PNF application rates. Compared with 0 N treatment, the amorphous region and single helix content of starch under high nitrogen (120 N and 180 N) treatments decreased by 2.07–8.06% and 15.17–20.82%, respectively, while the double helix content increased by 4.36–18.10%, all of which were significantly different. The crystal structure of starch was reflected by single helix and double helix, and the relative crystallinity of starch was elevated by topdressing PNF, which was consistent with the results of XRD analysis. In addition, the components of single helix and double helix showed opposite trends, which indicated that the formation mechanism of two helix structures differed. According to previous studies, the double helix structure was mainly composed of amylopectin side chains, and the single helix formation was associated with lipids and partly of amylose components [50,51], indicating that the decrease of amylose and crude fat content were the main reason for the decrease of single helix content.

### 3.5. Starch Solubility and Swelling Power

According to Appendix A, nitrogen, variety, and position had significant effect on starch solubility and swelling power. Figure 2 shows the changes in starch solubility and swelling power of japonica superior and inferior rice grains under different PNF treatments. The solubility of Nanjing 9108 superior grains was significantly lower than that of Nanjing 9108 inferior grains, and was comparable to that of inferior grains in Nanjing 0212 (Figure 4A,C). The swelling power of superior grains was 6.24–12.81% significantly lower than that of inferior grains in all PNF treatments (Figure 4B,D). Amylose limited the swelling of starch granules and maintained the integrity of swollen starch, and higher amylose content was the reason for the lower swelling power of superior grains. Starch granules with a high volume mean diameter had a large surface area, which tended to bind water easily, and had good hydrophilicity [52]. There was a certain number of differences in the starch granule size distribution between two varieties chosen in this study. The volume mean diameter of Nanjing 9108 superior grains was significantly lower than that of Nanjing 9108 inferior grains, while that of Nanjing 0212 superior grains was slightly higher than that of Nanjing 0212 inferior grains. This may explain the difference in solubility of superior and inferior grains.

Topdressing PNF improved the solubility and swelling power of starch. Compared with 0 N treatment, the solubility and swelling power of high nitrogen (120 N and 180 N) treatments increased by 10.35–22.55% and 6.06–14.91%, respectively, both reaching significance level (Figure 4). Furthermore, the starch solubility of Nanjing 0212 under topdressing PNF treatment was similar, and the starch solubility under high nitrogen treatments simply increased by 0.42–1.51% compared with the 60 N treatment, which indicated that higher amylose content could maintain the starch solubility. Amylose long chains could bind amylopectin clusters to limit starch granule swelling, suggesting that the increase in starch solubility and swelling power were associated with the decrease in amylose content [53]. In addition, the chain length distribution of amylopectin might also be a factor affecting the starch solubility and swelling power. The A chains and B1 chains in amylopectin would form a long double helix structure, which was difficult to dissociate at high temperature, thus restricting the swelling of starch granules [54]. In this study, the application of PNF decreased the content of A chain in amylopectin and increased the content of B2 chain and B3 chain, thereby elevating the starch solubility and swelling power.

### 3.6. Starch Thermal Properties

Nitrogen, variety, position, and N × V had significant effect on starch thermal properties, except for nitrogen on conclusion temperature (Appendix A). The differences in thermal properties of starch from superior and inferior grains of japonica rice under different PNF treatments are shown in Table 6, including gel properties and aging properties. The gelatinization enthalpy of superior grains was 1.15–5.80% lower than that of inferior grains, and the retrogradation enthalpy and percentage were 3.67–17.55% and 6.95–10.60% higher than those of inferior grains, respectively, both of which reached the significant level. The onset temperature (T_O_), the peak temperature (T_P_) and the conclusion temperature (T_C_) of superior grains were higher than those of inferior grains, and some reached a significant level. 

The gelatinization enthalpy reflected the energy required to dissociate the helix structure of starch, which was generally considered to be related to the amylose content and relative crystallinity. The retrogradation enthalpy and percentage reflected the recrystallization of gelatinized starch molecules after cooling. Amylose molecules crystallized rapidly during starch retrogradation [8]. Chung et al. [55] investigated the differences in thermal properties of rice varieties with different amylose content, and found that the gelatinization temperature increased along with amylose content. It followed that the difference in thermal properties of starch from superior and inferior grains was mainly related to the amylose content and relative crystallinity.

Topdressing PNF significantly increased the gelatinization enthalpy, and significantly decreased the retrogradation enthalpy and percentage (Table 6). The gelatinization enthalpy of high nitrogen (120 N and 180 N) treatments increased by 8.36–16.03% compared with 0 N treatment, and retrogradation enthalpy and percentage decreased by 5.96–19.97% and 17.18–27.98%, respectively. The changes of gelatinization temperature (T_O_, T_P_ and T_C_) under PNF treatments varied with varieties, showed an increasing trend in Nanjing 9108 but a decreasing trend in Nanjing 0212, and the difference became obvious with the increase of nitrogen application. Besides amylose content and relative crystallinity, amylopectin chain length distribution was another important factor affecting the thermal properties of starch. 

According to previous reports, long chains of amylopectin could form ordered crystal clusters, which had better heat resistance [56]. In this study, the gelatinization enthalpy increased with the increase of amylopectin B3 chain content, which indicated that the amylopectin B3 chain content was positively correlated with the gelatinization enthalpy. Although the amylose content of both cultivars was lower under high PNF treatments, the gelatinization temperature showed different trends, which indicated that the amylose content was not the dominant factor in the gelatinization temperature. The difference in gelatinization temperature might be related to the molecular structure of amylose. Li and Gong [57] found that the relative length of the amylose medium chain was negatively correlated with the range of gelatinization temperature, but the regulation of amylose molecular structure and gelatinization temperature as a result of nitrogen fertilizer application remains to be examined. The lipids existed in starch formed a single helix structure with amylose, which was difficult to dissociate. Gelatinization enthalpy increased with the decrease of the single helix content, while the retrogradation enthalpy showed the opposite trend, indicating that the single helix content was also related to starch thermal properties.

### 3.7. Starch Pasting Properties

According to Appendix A, nitrogen, variety, and position had significant effect on starch pasting properties, and some indicators had two-factor interactions and three-factor interactions significant effect. Table 7 displayed the RVA (Rapid Visco-Analyzer) characteristic parameters of starch from superior and inferior grains of japonica rice under different PNF treatments. The peak viscosity, hot viscosity and final viscosity of superior grains were significantly higher than those of inferior grains, and the difference became larger as the PNF application rates increased. Under 0 N treatment, the peak viscosity, hot viscosity and final viscosity of the superior grains were 14.99–21.04%, 15.99–24.95%, 11.71–16.30 higher than those of inferior grains, respectively, while under high nitrogen (120 N and 180 N) treatments, the parameters described above of superior grains were 28.54–45.72%, 39.14–64.68% and 31.63–36.31% higher than those of inferior grains, respectively, which indicated that an increase in PNF application rate was associated with a decrease in the pasting characteristics of inferior grains. Furthermore, the breakdown value of superior grains was higher than that of inferior grains, and the setback value was lower than that of inferior grains. The difference in pasting temperature of superior and inferior grains varied with varieties. The pasting temperature of superior grains of Nanjing 9108 was higher than that of inferior grains, while Nanjing 0212 showed the opposite trend. 

The breakdown value reflected the heat resistance of starch granules, and the setback value reflected the retrogradation of starch paste, which was related to the rearrangement of the leached amylose [58]. The disulfide bonds of proteins inhibited the swelling of starch granules and maintained the integrity of the swollen starch, which was not conducive to starch gelatinization, indicating that the lower content of total protein and protein component accounted for the better pasting characteristics of superior grains [59]. Large starch granules had a larger surface area and were more likely to bind water in order to gelatinize. The difference in pasting temperature of superior and inferior grains between the two varieties might be associated with the content of large starch granules.

With increasing nitrogen application, the peak viscosity, hot viscosity and final viscosity decreased significantly by 5.09–33.93%, 8.41–43.54% and 7.14–30.74%, respectively (Table 7). After topdressing PNF, the setback value decreased in the beginning and then increased, and the pasting temperature increased, while the breakdown value showed few differences. The setback value under 120 N treatment was significantly lower than that of 0 N treatment, and the pasting temperature of inferior grains was more susceptible to be regulated by PNF. The pasting temperature was the temperature at which starch granules swelled and broken when heated by water, and the viscosity began to rise rapidly [14]. Starch granules with many short amylopectin chains of amylopectin tended to swell and break, resulting in a decrease in the pasting temperature, which indicated that the increase in the pasting temperature was associated with the reduction of the A-chain content of amylopectin by PNF [60]. 

The pasting characteristics comprehensively reflected the swelling, disintegration, and gelation ability of starch granules. Hu et al. [61] found that nitrogen fertilizer decreased the peak viscosity, hot viscosity, and final viscosity, but had few effects on breakdown and setback values, which was consistent with our study. Lu et al. [26] also found that topdressing nitrogen fertilizer at heading stage decreased pasting viscosity of starch, and resulted in a decrease in the eating quality. Previous studies argued that the higher the proportion of long chains in amylopectin, the stronger the interaction between starch molecules, and the easier it was to maintain the stability and integrity of starch granules [62]. In this study, content of both total protein and protein components increased along with PNF application rates, and so were the lengths of amylopectin B3 chain and average chain, which led to the reduction of peak viscosity, hot viscosity and final viscosity.

### 3.8. Instrumental Measurement of Eating Quality

#### 3.8.1. Taste Value Measurement by NIR

Nitrogen, variety, position, and two-factor interactions had significant effect on taste value (Appendix A). Figure 5 showed the variations of taste value, hardness, and stickiness of superior and inferior grains of japonica rice under different rates of PNF. The taste value was determined by the Satake taste meter, which evaluated the appearance, hardness, viscosity, and balance (ratio of viscosity to hardness) of the rice based on the near-infrared method was used to obtain the taste value. The taste value of superior grains was higher than that of inferior grains, and the difference was especially significant in Nanjing 0212, whose taste value of superior grains was 2.76–12.40% higher than that of inferior grains. (Figure 5A,D). Previous studies showed that taste value was negatively correlated with protein content, but positively correlated with breakdown value [26]. This indicated that the lower protein content and higher breakdown value were conducive to the good taste value of superior grains. In addition, the relative crystallinity and single helix content of the superior grains were relatively lower, so the starch was enabled to consume less energy to dissociate the helical structure, which meant lower gelatinization enthalpy, thus exhibiting better cooking and eating quality under the same cooking condition. 

Topdressing PNF reduced the taste value of superior and inferior grains of japonica rice: the taste values of superior grains of Nanjing 9108 and Nanjing 0212 decreased by 0.72–4.40% and 2.90–11.47%, respectively, and the inferior grains decreased by 2.95–11.47% and 5.35–18.32%, respectively. This indicated that the low taste value varieties and inferior grains were more inclined to be regulated by PNF, which meant that the eating quality of inferior grains could be optimized by selecting varieties with good taste and reducing the application of PNF during cultivation. It was generally believed that the amylose content was closely related to the cooking and eating quality, but we found that the taste value was significantly different with similar amylose content in some treatments. We infer that this was related to the amylopectin chain length distribution and starch thermal characteristics. PNF reduced the proportion of amylopectin A chain and increased the proportion of B2 chain and B3 chain, which increased the double helix content and made starch crystallization more stable, thereby limiting the swelling of starch granules, leading to an increase in the gelatinization enthalpy, and ultimately a deterioration in the eating quality.

#### 3.8.2. Texture Analysis

According to Appendix A, hardness was significant affected by nitrogen and position, and stickiness was significant affected by nitrogen, variety, N × P, and V × P. The differences in texture between superior and inferior grains of japonica rice were further determined by texture analyzer, in which the stickiness value was negative, and its absolute value reflected the stickiness of rice. The results showed that the hardness of superior grains was 9.80–34.26% significantly higher than that of inferior grains (Figure 5B,E), and the difference in the stickiness of superior grains and inferior grains varied with the varieties. The stickiness of superior grains in Nanjing 9108 was higher than that of inferior grains, while the trend of Nanjing 0212 was opposite (Figure 5C,F). The difference in hardness was related to the amylose content, while the difference in stickiness might be related to the molecular weight of amylopectin. Amylopectin with small molecular weight tended to be leached and attached to the surface of rice grains during cooking, thus increasing the stickiness [10,23]. 

Topdressing PNF increased the hardness and stickiness of superior and inferior grains of japonica rice, and the hardness and stickiness of high nitrogen (120 N and 180 N) treatments increased by 16.27–48.40% and 5.30–19.09% compared with 0 N treatment, reaching a significant level of difference. Proteins, especially prolamins, provided support for rice grains during cooking [63]. Previous study found that rice with high content of long branch-chain in amylopectin was less susceptible to swell during cooking, which had stronger intermolecular forces and higher stickiness [17]. In conclusion, although PNF reduced the amylose content and facilitated the swelling of starch granules, the increase of protein and prolamin counteracted this promotion, resulting in higher hardness. The increase in stickiness was mainly related to the increase of amylopectin B3 chains.

## 4. Conclusions

The results showed that, compared with inferior grains, the superior grains had better pastings properties and high taste value, namely better cooking and eating quality, which was related to the differences in biochemical components and starch crystal structure between superior and inferior grains. Under all nitrogen levels, superior grains had higher amylose content, lower protein and crude fat content than inferior grains, which contributed to the decrease of relative crystallinity and single helix content of starch, making starch granules susceptible to swell during cooking and retrograde after cooling, showing better cooking and eating quality as a whole. PNF deteriorated the cooking and eating quality of superior and inferior grains of japonica rice, which increased hardness, decreased taste value and worsened pasting characteristics. This was related to the fact that PNF increased protein content and decreased starch and crude fat content. However, we also found that there was significant difference in taste value of rice with few differences in starch and crude fat content among some treatments. This meant that the regulation of PNF on physicochemical properties and fine structure of starch was another essential factor in the deterioration of cooking and eating quality. Topdressing PNF reduced the content and volume mean diameter of large starch granules, increased the average chain length of amylopectin and starch relative crystallinity, which limited starch swelling and gelatinization, resulting in increased gelatinization enthalpy and decreased pasting viscosity, thus leading to deterioration of cooking and eating quality, and this effect was more evident under high nitrogen treatments. Moreover, with the increase of PNF, the cooking and eating quality of low taste varieties and inferior grains deteriorated significantly, which suggested that the quality of inferior grains could be optimized by variety selection and reduction of panicle fertilizer in production. We suggest that rice growers improve the eating quality of rice by selecting varieties with good taste and topdressing PNF with 60 kg N ha^−1^, so as to meet consumers’ demand for rice palatability. These findings enabled us to better understand the differences in cooking and eating quality of superior and inferior grains and their relationships with starch properties and structure, and to clarify the response of starch physicochemical properties and fine structure to PNF. New ideas were put forward for cultivation measures to optimize rice eating quality and starch functional properties.

## Figures and Tables

**Figure 1 foods-11-02489-f001:**
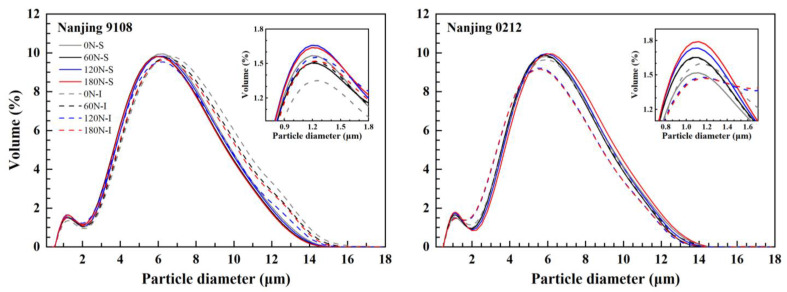
Starch granule size distributions of superior and inferior grains of japonica rice under different PNF treatments.

**Figure 2 foods-11-02489-f002:**
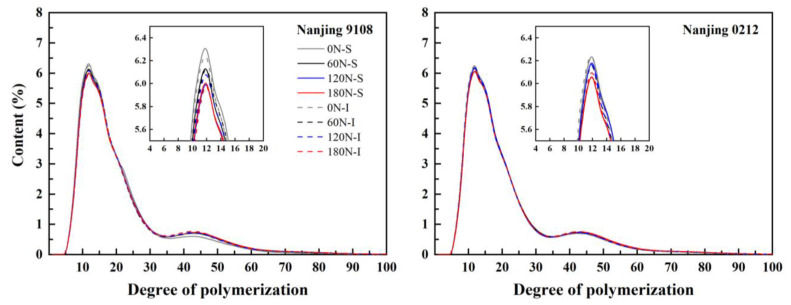
Amylopectin chain length distributions of superior and inferior grains of japonica rice under different PNF treatments.

**Figure 3 foods-11-02489-f003:**
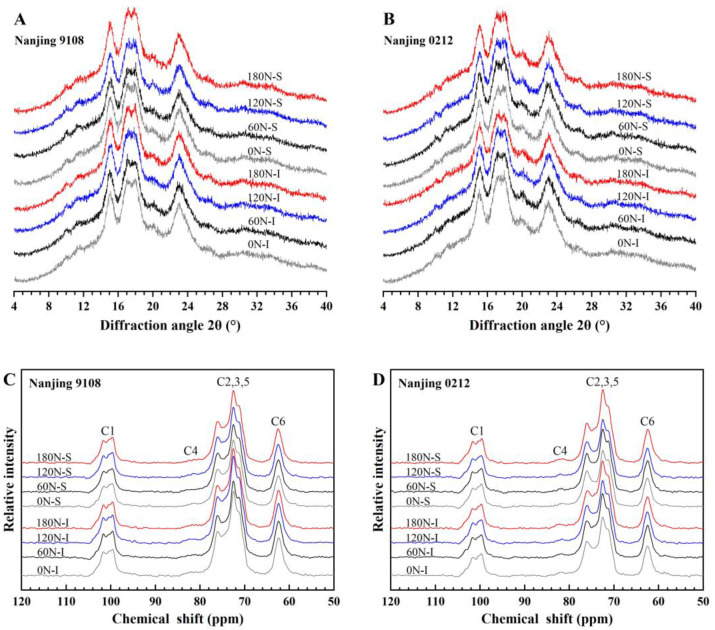
XRD patterns (**A**,**B**) and NMR ordered sub-spectra (**C**,**D**) of superior and inferior grains of japonica rice under different PNF treatments.

**Figure 4 foods-11-02489-f004:**
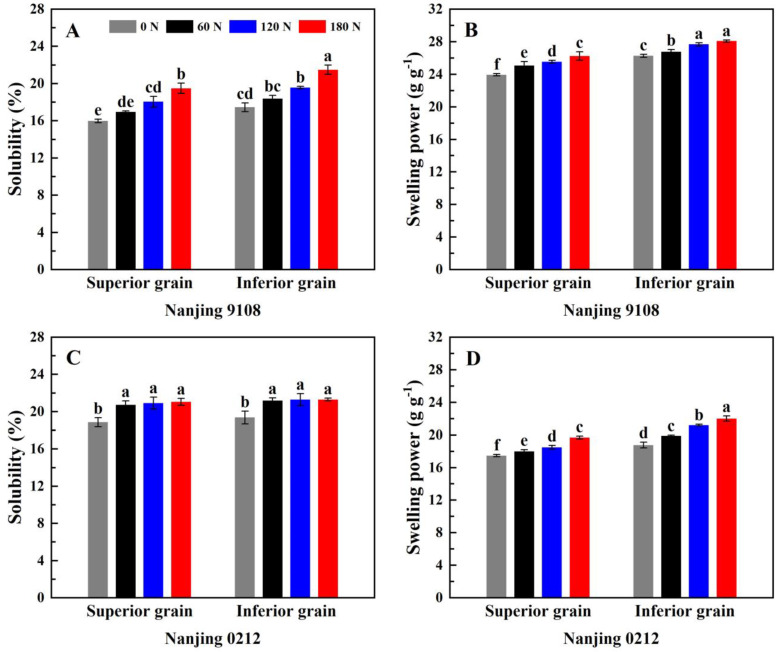
Solubility (**A**,**C**) and swelling power (**B**,**D**) of superior and inferior grains of japonica rice under different PNF treatments. Values with different letters are significantly different (*p* < 0.05).

**Figure 5 foods-11-02489-f005:**
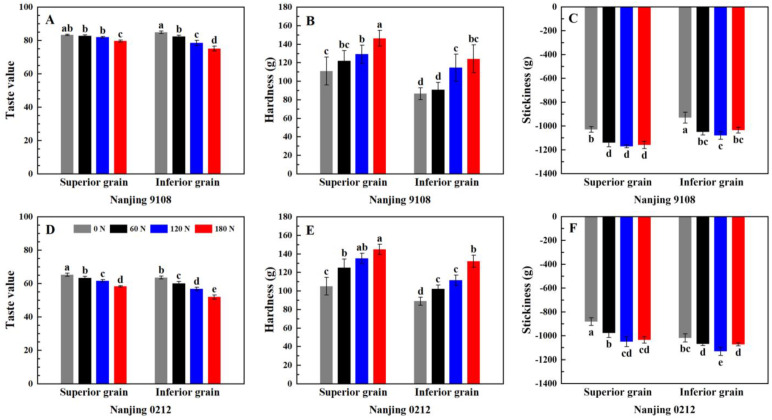
Taste value (**A**,**D**), hardness (**B**,**E**) and stickiness (**C**,**F**) of superior and inferior grains of japonica rice under different PNF treatments. Values with different letters are significantly different (*p* < 0.05).

**Table 1 foods-11-02489-t001:** Contents of total starch, amylose and amylopectin in superior and inferior grains of japonica rice under different PNF treatments.

Variety	Position	PNF (kg ha^−1^)	Starch Content (%)	Amylose Content (%)	Amylopectin Content (%)	Amylose/Amylopectin
Nanjing 9108	Superior grain	0	82.91 ± 1.23 b	11.86 ± 0.07 a	70.76 ± 0.93 c	0.168 ± 0.002 a
		60	81.48 ± 1.34 b	11.53 ± 0.20 a	69.53 ± 0.50 de	0.166 ± 0.001 a
		120	79.07 ± 0.97 cd	10.35 ± 0.11 b	68.55 ± 0.92 e	0.151 ± 0.002 b
		180	75.46 ± 0.59 e	9.35 ± 0.38 cd	66.00 ± 0.64 f	0.142 ± 0.001 c
	Inferior grain	0	85.27 ± 1.53 a	10.37 ± 0.33 b	75.44 ± 0.74 a	0.138 ± 0.001 c
		60	82.58 ± 1.35 b	9.75 ± 0.08 bc	72.66 ± 1.23 b	0.134 ± 0.002 d
		120	79.75 ± 0.70 c	9.39 ± 0.15 cd	70.48 ± 0.67 cd	0.133 ± 0.001 d
		180	77.96 ± 0.77 d	8.92 ± 0.16 d	68.79 ± 0.32 e	0.130 ± 0.001 d
Nanjing 0212	Superior grain	0	83.97 ± 0.57 b	20.86 ± 0.27 a	63.14 ± 0.47 e	0.330 ± 0.002 a
		60	81.84 ± 0.29 c	20.12 ± 0.36 b	61.85 ± 0.38 f	0.325 ± 0.002 b
		120	81.08 ± 0.34 cd	19.53 ± 0.23 b	61.47 ± 0.13 f	0.318 ± 0.001 c
		180	79.91 ± 0.78 d	18.79 ± 0.55 c	61.00 ± 0.89 f	0.308 ± 0.004 d
	Inferior grain	0	86.67 ± 0.44 a	16.69 ± 0.17 d	70.05 ± 0.42 a	0.238 ± 0.001 e
		60	83.76 ± 0.44 b	16.08 ± 0.12 d	67.78 ± 0.35 b	0.237 ± 0.001 e
		120	82.20 ± 0.81 c	15.29 ± 0.04 e	66.73 ± 0.64 c	0.229 ± 0.002 f
		180	80.23 ± 0.93 d	14.49 ± 0.56 f	65.24 ± 0.26 d	0.222 ± 0.001 g

Data are expressed as the mean ± SD (*n* = 3). Values in the same varieties and column with different letters are significantly different (*p* < 0.05). PNF: panicle nitrogen fertilizer.

**Table 2 foods-11-02489-t002:** Contents of total protein, protein components and crude fat in superior and inferior grains of japonica rice under different PNF treatments.

Variety	Position	PNF (kg ha^−1^)	Total Protein Content (%)	Albumin Content (mg g^−1^)	Globulin Content (mg g^−1^)	Prolamin Content (mg g^−1^)	Glutelin Content (mg g^−1^)	Crude Fat Content (%)
Nanjing 9108	Superior grain	0	5.94 ± 0.03 h	3.45 ± 0.10 f	4.32 ± 0.04 d	4.10 ± 0.02 f	33.44 ± 0.27 f	0.65 ± 0.02 c
		60	6.64 ± 0.05 f	3.57 ± 0.08 f	4.44 ± 0.05 d	4.51 ± 0.04 d	38.56 ± 1.42 e	0.45 ± 0.01 d
		120	7.04 ± 0.01 e	3.77 ± 0.09 e	4.61 ± 0.05 c	4.86 ± 0.07 c	42.09 ± 0.75 d	0.42 ± 0.04 d
		180	7.19 ± 0.03 d	4.03 ± 0.05 d	4.74 ± 0.06 c	5.35 ± 0.03 b	43.75 ± 0.98 cd	0.42 ± 0.02 d
	Inferior grain	0	6.54 ± 0.02 g	4.39 ± 0.16 c	4.41 ± 0.04 d	4.57 ± 0.05 e	35.03 ± 0.39 f	1.28 ± 0.03 a
		60	7.49 ± 0.06 c	4.52 ± 0.05 bc	4.71 ± 0.06 c	4.67 ± 0.03 cd	45.49 ± 0.67 bc	0.84 ± 0.04 b
		120	7.90 ± 0.07 b	4.66 ± 0.07 b	4.89 ± 0.07 b	4.98 ± 0.02 bc	46.27 ± 0.46 b	0.83 ± 0.03 b
		180	8.29 ± 0.00 a	4.90 ± 0.10 a	5.09 ± 0.05 a	5.43 ± 0.06 a	48.68 ± 0.72 a	0.82 ± 0.02 b
Nanjing 0212	Superior grain	0	5.69 ± 0.01 h	2.97 ± 0.06 d	4.16 ± 0.04 e	4.12 ± 0.04 f	37.04 ± 0.70 e	0.52 ± 0.03 d
		60	6.30 ± 0.00 f	3.13 ± 0.06 d	4.28 ± 0.06 de	4.80 ± 0.10 e	37.53 ± 0.51 e	0.39 ± 0.03 e
		120	6.62 ± 0.01 e	3.42 ± 0.08 c	4.40 ± 0.08 d	5.02 ± 0.05 d	41.44 ± 0.31 c	0.34 ± 0.03 f
		180	6.85 ± 0.01 d	3.72 ± 0.16 b	4.71 ± 0.10 b	5.22 ± 0.10 a	44.94 ± 0.90 b	0.33 ± 0.02 f
	Inferior grain	0	6.05 ± 0.00 g	3.63 ± 0.05 b	4.55 ± 0.03 c	4.47 ± 0.04 f	38.54 ± 0.69 de	1.79 ± 0.02 a
		60	7.59 ± 0.00 c	3.75 ± 0.08 b	4.57 ± 0.11 c	4.94 ± 0.06 d	39.92 ± 0.30 cd	1.28 ± 0.02 b
		120	8.44 ± 0.01 b	3.97 ± 0.10 a	5.04 ± 0.05 a	5.09 ± 0.09 c	48.38 ± 0.58 a	1.18 ± 0.03 c
		180	8.58 ± 0.04 a	4.12 ± 0.08 a	5.13 ± 0.09 a	5.48 ± 0.06 a	49.88 ± 1.00 a	1.16 ± 0.02 c

Data are expressed as the mean ± SD (*n* = 3). Values in the same varieties and column with different letters are significantly different (*p* < 0.05). PNF: panicle nitrogen fertilizer.

**Table 3 foods-11-02489-t003:** Starch granule size distributions of superior and inferior grains of japonica rice under different PNF treatments.

Variety	Position	PNF (kg ha^−1^)	Small Starch Granules ≤2 μm (%)	Middle Starch Granules 2–5 μm (%)	Large Starch Granules ≥5 μm (%)	Volume Mean Diameter (μm)
Nanjing 9108	Superior grain	0	12.72 ± 0.03 c	34.11 ± 0.06 e	53.17 ± 0.09 d	5.33 ± 0.01 d
		60	12.60 ± 0.05 c	36.31 ± 0.01 a	51.10 ± 0.06 g	5.22 ± 0.00 g
		120	13.47 ± 0.11 a	34.26 ± 0.02 d	52.27 ± 0.13 e	5.27 ± 0.01 f
		180	13.56 ± 0.12 a	35.89 ± 0.13 b	50.55 ± 0.25 h	5.17 ± 0.02 h
	Inferior grain	0	10.97 ± 0.08 e	30.04 ± 0.04 h	58.99 ± 0.13 a	5.78 ± 0.01 a
		60	12.29 ± 0.09 d	31.28 ± 0.05 g	56.43 ± 0.13 b	5.58 ± 0.01 b
		120	13.14 ± 0.12 b	34.84 ± 0.04 c	52.02 ± 0.15 f	5.31 ± 0.01 e
		180	12.59 ± 0.14 c	32.26 ± 0.04 f	55.15 ± 0.17 c	5.50 ± 0.01 c
Nanjing 0212	Superior grain	0	13.54 ± 0.11 d	34.51 ± 0.08 h	51.95 ± 0.11 a	5.22 ± 0.01 a
		60	14.01 ± 0.07 c	36.41 ± 0.02 g	49.58 ± 0.06 b	5.07 ± 0.00 b
		120	14.22 ± 0.06 b	37.80 ± 0.05 d	47.98 ± 0.11 e	4.99 ± 0.01 c
		180	14.45 ± 0.02 a	36.57 ± 0.03 f	48.98 ± 0.01 c	5.06 ± 0.00 b
	Inferior grain	0	13.96 ± 0.02 c	37.37 ± 0.04 e	48.67 ± 0.05 d	5.06 ± 0.00 b
		60	14.09 ± 0.17 bc	38.82 ± 0.03 c	47.09 ± 0.16 f	4.99 ± 0.01 c
		120	14.40 ± 0.06 a	42.32 ± 0.03 b	43.28 ± 0.06 g	4.77 ± 0.00 d
		180	14.44 ± 0.02 a	42.53 ± 0.05 a	43.02 ± 0.07 h	4.77 ± 0.01 d

Data are expressed as the mean ± SD (*n* = 3). Values in the same varieties and column with different letters are significantly different (*p* < 0.05). PNF: panicle nitrogen fertilizer.

**Table 4 foods-11-02489-t004:** Amylopectin chain length distributions of superior and inferior grains of japonica rice under different PNF treatments.

Variety	Position	PNF (kg ha^−1^)	A Chain (%)	B1 Chain (%)	B2 Chain (%)	B3 Chain (%)	(A + B1)/(B2 + B3)	Amylopectin Average DP
Nanjing 9108	Superior grain	0	28.60 ± 0.10 a	47.21 ± 0.21 a	10.83 ± 0.08 c	13.36 ± 0.04 e	3.13 ± 0.01 a	20.76 ± 0.05 e
		60	27.16 ± 0.13 c	46.82 ± 0.22 a	11.36 ± 0.13 ab	14.66 ± 0.06 cd	2.84 ± 0.02 bc	21.61 ± 0.05 c
		120	26.26 ± 0.12 f	46.60 ± 0.21 a	11.53 ± 0.14 a	15.61 ± 0.19 ab	2.72 ± 0.01 d	22.12 ± 0.07 a
		180	26.51 ± 0.20 ef	46.50 ± 0.16 a	11.28 ± 0.03 ab	15.76 ± 0.20 a	2.70 ± 0.01 d	22.12 ± 0.12 a
	Inferior grain	0	28.19 ± 0.10 b	46.43 ± 0.16 a	10.93 ± 0.04 c	14.46 ± 0.11 d	2.94 ± 0.01 b	21.40 ± 0.14 d
		60	26.94 ± 0.02 cd	46.98 ± 0.04 a	10.96 ± 0.12 c	15.12 ± 0.06 bc	2.83 ± 0.01 bc	21.80 ± 0.01 bc
		120	26.70 ± 0.03 de	46.96 ± 0.24 a	11.11 ± 0.04 bc	15.23 ± 0.15 ab	2.80 ± 0.03 cd	21.84 ± 0.13 b
		180	26.53 ± 0.15 ef	47.01 ± 0.11 a	11.08 ± 0.18 bc	15.39 ± 0.22 ab	2.78 ± 0.03 cd	21.93 ± 0.11 ab
Nanjing 0212	Superior grain	0	27.26 ± 0.07 b	47.06 ± 0.25 a	10.92 ± 0.16 ab	14.76 ± 0.12 cd	2.89 ± 0.02 ab	21.62 ± 0.07 d
		60	26.71 ± 0.02 cd	47.06 ± 0.18 a	11.01 ± 0.12 a	15.21 ± 0.15 bc	2.81 ± 0.02 bc	21.86 ± 0.10 bc
		120	26.28 ± 0.07 e	47.02 ± 0.19 a	11.08 ± 0.06 a	15.61 ± 0.14 ab	2.75 ± 0.01 c	22.05 ± 0.08 ab
		180	26.36 ± 0.08 e	46.82 ± 0.04 a	11.04 ± 0.03 a	15.78 ± 0.05 a	2.73 ± 0.01 c	22.13 ± 0.04 a
	Inferior grain	0	27.85 ± 0.15 a	47.01 ± 0.07 a	10.68 ± 0.08 b	14.51 ± 0.19 d	2.97 ± 0.04 a	21.53 ± 0.03 d
		60	26.77 ± 0.04 c	47.17 ± 0.23 a	10.96 ± 0.05 ab	15.10 ± 0.14 bc	2.84 ± 0.02 bc	21.71 ± 0.07 cd
		120	26.88 ± 0.01 c	46.80 ± 0.13 a	10.88 ± 0.08 ab	15.44 ± 0.21 ab	2.80 ± 0.02 bc	21.90 ± 0.10 bc
		180	26.44 ± 0.08 de	47.03 ± 0.13 a	11.02 ± 0.02 a	15.52 ± 0.10 ab	2.76 ± 0.01 c	21.94 ± 0.09 ab

Data are expressed as the mean ± SD (*n* = 3). Values in the same varieties and column with different letters are significantly different (*p* < 0.05). PNF: panicle nitrogen fertilizer.

**Table 5 foods-11-02489-t005:** Relative crystallinity, amorphous region, contents of single helix and double helix of superior and inferior grains of japonica rice under different PNF treatments.

Variety	Position	PNF (kg ha^−1^)	Relative Crystallinity (%)	Amorphous (%)	Single Helix (%)	Double Helix (%)
Nanjing 9108	Superior grain	0	24.72 ± 0.05 d	57.04 ± 0.40 a	3.86 ± 0.12 bc	39.10 ± 0.49 e
		60	24.82 ± 0.04 d	56.81 ± 0.31 a	3.82 ± 0.06 bc	39.37 ± 0.25 e
		120	24.91 ± 0.04 d	56.56 ± 0.12 a	3.07 ± 0.08 d	40.37 ± 0.07 d
		180	25.33 ± 0.11 c	55.82 ± 0.14 b	2.90 ± 0.06 d	41.28 ± 0.20 c
	Inferior grain	0	24.92 ± 0.14 d	54.63 ± 0.45 c	4.41 ± 0.09 a	40.96 ± 0.36 cd
		60	25.74 ± 0.15 b	54.41 ± 0.21 c	4.12 ± 0.05 ab	41.47 ± 0.21 c
		120	25.99 ± 0.04 ab	52.82 ± 0.10 d	3.98 ± 0.24 bc	43.19 ± 0.17 b
		180	26.14 ± 0.17 a	52.26 ± 0.35 d	3.68 ± 0.01 c	44.06 ± 0.35 a
Nanjing 0212	Superior grain	0	23.35 ± 0.16 f	64.27 ± 0.07 a	3.53 ± 0.13 b	32.21 ± 0.14 f
		60	23.57 ± 0.11 ef	63.20 ± 0.15 b	3.34 ± 0.09 b	33.46 ± 0.17 e
		120	23.99 ± 0.07 cd	62.68 ± 0.12 b	3.01 ± 0.35 c	34.32 ± 0.39 d
		180	24.46 ± 0.11 b	59.18 ± 0.56 d	2.92 ± 0.15 c	37.90 ± 0.41 b
	Inferior grain	0	23.82 ± 0.19 de	63.02 ± 0.22 b	4.23 ± 0.34 a	32.74 ± 0.48 f
		60	24.27 ± 0.12 bc	62.78 ± 0.07 b	3.59 ± 0.09 b	33.64 ± 0.08 e
		120	24.90 ± 0.09 a	61.26 ± 0.25 c	3.42 ± 0.41 b	35.32 ± 0.31 c
		180	25.27 ± 0.15 a	57.86 ± 0.32 e	3.34 ± 0.07 b	38.81 ± 0.36 a

Data are expressed as the mean ± SD (*n* = 3). Values in the same varieties and column with different letters are significantly different (*p* < 0.05). PNF: panicle nitrogen fertilizer.

**Table 6 foods-11-02489-t006:** Starch thermal properties of superior and inferior grains of japonica rice under different PNF treatments.

Variety	Position	PNF (kg ha^−1^)	△H_gel_ (J·g^−1^)	T_O_ (°C)	T_P_ (°C)	T_C_ (°C)	△H_ret_ (J·g^−1^)	R (%)
Nanjing 9108	Superior grain	0	10.20 ± 0.07 e	62.80 ± 0.17 c	70.33 ± 0.15 bc	77.53 ± 0.12 bc	2.83 ± 0.03 a	27.75 ± 0.49 a
		60	11.67 ± 0.03 c	63.37 ± 0.15 b	70.57 ± 0.06 ab	77.80 ± 0.20 ab	2.72 ± 0.04 b	23.31 ± 0.35 c
		120	12.00 ± 0.03 b	63.73 ± 0.12 a	70.70 ± 0.17 a	77.93 ± 0.21 a	2.69 ± 0.08 b	22.44 ± 0.62 d
		180	11.73 ± 0.06 c	63.80 ± 0.20 a	70.70 ± 0.20 a	78.00 ± 0.26 a	2.56 ± 0.03 c	21.85 ± 0.28 de
	Inferior grain	0	10.61 ± 0.05 d	59.63 ± 0.32 e	69.93 ± 0.15 d	77.23 ± 0.15 c	2.72 ± 0.05 b	25.67 ± 0.37 b
		60	11.94 ± 0.08 b	62.17 ± 0.15 d	70.17 ± 0.15 cd	77.53 ± 0.21 bc	2.58 ± 0.06 c	21.63 ± 0.44 e
		120	12.14 ± 0.04 a	62.53 ± 0.15 c	70.43 ± 0.06 b	77.67 ± 0.32 ab	2.53 ± 0.04 c	20.84 ± 0.28 f
		180	11.95 ± 0.08 b	62.57 ± 0.12 c	70.50 ± 0.10 ab	77.77 ± 0.21 ab	2.38 ± 0.01 d	19.94 ± 0.19 g
Nanjing 0212	Superior grain	0	10.71 ± 0.05 f	63.47 ± 0.12 a	70.60 ± 0.10 a	77.63 ± 0.12 a	3.20 ± 0.08 a	29.92 ± 0.61 a
		60	11.31 ± 0.08 e	63.23 ± 0.06 ab	70.43 ± 0.12 a	77.37 ± 0.06 ab	2.95 ± 0.05 c	26.06 ± 0.54 c
		120	11.84 ± 0.08 c	63.10 ± 0.10 bc	70.17 ± 0.06 b	77.23 ± 0.25 b	2.90 ± 0.06 cd	24.50 ± 0.69 d
		180	11.87 ± 0.05 c	61.93 ± 0.21 d	69.20 ± 0.30 d	76.83 ± 0.25 c	2.57 ± 0.03 f	21.66 ± 0.36 f
	Inferior grain	0	11.31 ± 0.07 e	62.90 ± 0.10 c	69.57 ± 0.06 c	76.63 ± 0.15 cd	3.09 ± 0.04 b	27.33 ± 0.44 b
		60	11.61 ± 0.08 d	61.27 ± 0.21 e	69.03 ± 0.25 de	76.30 ± 0.20 de	2.83 ± 0.05 d	24.37 ± 0.46 d
		120	12.00 ± 0.07 b	61.23 ± 0.12 e	69.00 ± 0.00 de	76.17 ± 0.06 e	2.75 ± 0.05 e	22.89 ± 0.51 e
		180	12.60 ± 0.05 a	61.00 ± 0.10 e	68.90 ± 0.10 e	75.97 ± 0.15 e	2.47 ± 0.01 g	19.58 ± 0.17 g

Data are expressed as the mean ± SD (*n* = 3). Values in the same varieties and column with different letters are significantly different (*p* < 0.05). PNF: panicle nitrogen fertilizer. ΔH_gel_, gelatinization enthalpy; T_O_, onset temperature; T_P_, peak of gelatinization temperature; T_C_, conclusion temperature; ΔH_ret_, retrogradation enthalpies; R, retrogradation percentage.

**Table 7 foods-11-02489-t007:** Starch pasting properties of rice flour from superior and inferior grains of japonica rice under different PNF treatments.

Variety	Position	PNF (kg ha^−1^)	Peak Viscosity (cP)	Hot Viscosity (cP)	Breakdown (cP)	Final Viscosity (cP)	Setback (cP)	Pasting Temperature (°C)
Nanjing 9108	Superior grain	0	3776 ± 11 a	2509 ± 12 a	1267 ± 14 a	3003± 9 a	−773 ± 19 d	76.33 ± 0.31 a
		60	3584 ± 27 b	2298 ± 30 b	1286 ± 18 a	2789 ± 43 b	−795 ± 21 de	76.80 ± 0.05 a
		120	3566 ± 30 bc	2274 ± 49 b	1292 ± 28 a	2721 ± 33 c	−844 ± 29 f	76.78 ± 0.06 a
		180	3518 ± 15 c	2267 ± 24 b	1251 ± 14 a	2701 ± 17 c	−817 ± 13 ef	76.80 ± 0.05 a
	Inferior grain	0	3120 ± 12 d	2008 ± 32 c	1112 ± 21 bc	2582 ± 51 d	−537 ± 38 a	75.52 ± 0.35 b
		60	2956 ± 28 e	1822 ± 11 d	1133 ± 18 b	2264 ± 15 e	−692 ± 15 c	76.30 ± 0.38 a
		120	2774 ± 11 f	1634 ± 31 e	1140 ± 27 b	2067 ± 30 f	−707 ± 20 c	76.62 ± 0.28 a
		180	2612 ± 15 g	1523 ± 27 f	1089 ± 26 c	1988 ± 51 g	−624 ± 37 b	76.70 ± 0.09 a
Nanjing 0212	Superior grain	0	3478 ± 11 a	2769 ± 20 a	710 ± 13 a	3794 ± 24 a	316 ± 14 ab	77.38 ± 0.03 d
		60	3212 ± 41 b	2491 ± 51 b	721 ± 14 a	3475 ± 30 b	263 ± 12 cd	77.60 ± 0.48 d
		120	2992 ± 26 c	2266 ± 41 d	726 ± 18 a	3228 ± 43 d	237 ± 17 d	77.73 ± 0.33 cd
		180	2925 ± 24 d	2219 ± 25 d	706 ± 16 a	3207 ± 12 d	282 ± 12 bc	77.67 ± 0.45 d
	Inferior grain	0	3038 ± 11 c	2387 ± 28 c	651 ± 33 b	3397 ± 19 c	359 ± 10 a	78.18 ± 0.03 bc
		60	2360 ± 61 e	1671 ± 66 e	689 ± 30 ab	2677 ± 63 e	317 ± 11 ab	78.60 ± 0.09 b
		120	2179 ± 47 f	1471 ± 32 f	708 ± 16 a	2478 ± 37 f	298 ± 13 bc	79.43 ± 0.46 a
		180	2007 ± 32 g	1348 ± 20 g	660 ± 14 b	2353 ± 24 g	345 ± 11 a	79.70 ± 0.05 a

Data are expressed as the mean ± SD (*n* = 3). Values in the same varieties and column with different letters are significantly different (*p* < 0.05). PNF: panicle nitrogen fertilizer.

## Data Availability

The data presented in this study are available on request from the corresponding author.

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
