# Peer review of "The Starch Physicochemical Properties between Superior and Inferior Grains of Japonica Rice under Panicle Nitrogen Fertilizer Determine the Difference in Eating Quality"

_foods, 2022, doi:10.3390/foods11162489_

Round 1
Reviewer 1 Report
1. In the Introduction = please add more information about the standard of good palatability of rice for rice consumers (related to the starch physicochemical properties, fine structure, and texture properties).
2. In the Materials and Methods = please explain detailly about the application of Nitrogen fertilizer at the panicle initiation and spikelet differentiation stages, especially related to the 2 rice genotypes that you use: Nanjing 9018 and Nanjing 0212, are they have panicle initiation and spikelet differentiation stages at the same time, so you applied the Nitrogen fertilizer as the same time?
3. In the Discussion = it is better to add more information about comparison of application Nitrogen fertilizer at the panicle initiation and spikelet differentiation stages (your results) with application Nitrogen fertilizer at the other growth stages (from previous studies) related to the starch physicochemical properties, fine structure, and texture properties.
4. In the Conclusions = please give suggestions how to increase the starch physicochemical properties, fine structure, and texture properties of rice to full fill the rice consumer demands.
Author Response
Dear review experts,
Thank you for your letter and the comments concerning our manuscript. We have carefully reviewed and revised the manuscript. All revisions to the manuscript were marked up with the “Track Changes” function. Please refer to the attachment, and we hope you will be satisfied.
We hope that the revised manuscript can meet your requirement and can be published in the Foods. Thank you again for your professional review work on our manuscript. Finally, give my most sincere wishes for you.
Sincerely yours,
Yan Jiang
College of Agriculture, Yangzhou University
Yangzhou 225009, Jiangsu, China

Reviewer 2 Report
English usage: in places the English usage is problematic. I've gone through and suggested changes in some places, but the whole thing needs to be reviewed. The latter part of the results and discussion is better in that regard.
Writing style: the use of large paragraphs makes the content hard to follow than it needs to be. Break them up. I've made suggestions for this in some paragraphs, but by no means all.
For example, consistently follow the structure of an introductory few sentences, then a paragraph on variety, a paragraph on topdressing, and a paragraph on inferior/superior grain.
You have generally done this, but within the big paragraphs. Breaking it up means that it will be clearer where to put things. For example, the comments about Crude Fat.
Sections: think about breaking up some of the sections. For example, "3.8 Taste value and textural properties" could be renamed "3.8 Instrumental measurement of eating quality", and then have sub-sections "3.8.1 Taste value measurement by NIR", and "3.8.2 Texture analysis".
Satake taste meter: did you only do the taste value measurement? Or did you include freshness, hardness, stickiness, ?
Statistics: some sort of information about the significance of the main effects and interactions is needed.
Was the ANOVA: (rice variety; 2 levels) x (superior or inferior grain; 2 levels) x (N addition rate; 4 levels) x (replication via different plots; not sure, is it 3?)?
If so, are all the main effects, two way interactions and three way interactions significant for every test? If this is the case, then say this.
Otherwise report what was significant in terms of main effects and interactions. I know this will be large as there are many tests performed, but without it understanding what is going on isn't possible.
It could go into the supplementary material.

Author Response
Dear review experts,
Thank you for your letter and the comments concerning our manuscript. We have carefully reviewed and revised the manuscript. All revisions to the manuscript were marked up with the “Track Changes” function. As for the questions you marked in pdf file, we replied in the revised manuscript. Please refer to the attachment, and we hope you will be satisfied. If you there still have problems with our English usage, please let us know, we will consider using the editing services provided by MDPI to make up for our shortcomings.
We hope that the revised manuscript can meet your requirement and can be published in the Foods. Thank you again for your professional review work on our manuscript. Finally, give my most sincere wishes for you.
Sincerely yours,
Yan Jiang
College of Agriculture, Yangzhou University
Yangzhou 225009, Jiangsu, China

Round 2
Reviewer 2 Report
Dear Authors
Thanks for the revised manuscript. I can see the work that has been done, such as insertion of paragraphs and sections. Particularly, thank you for the ANOVA results.
There are still some areas to work on but it is getting close.
Please go through the attached for my detailed comments.

Author Response
Dear review experts,
Thank you so much for your careful check and valuable comments concerning our manuscript. We have carefully reviewed and revised the manuscript. All revisions to the manuscript were marked up with the “Track Changes” function. As for the questions you marked in pdf file, we replied in the revised manuscript. We hope you will be satisfied.
We hope that the revised manuscript can meet your requirement and can be published in the Foods. If you still have questions about phosphorus and potassium fertilizer, please refer to the attachment. Thank you again for your professional review work on our manuscript. Finally, give my most sincere wishes for you.
Sincerely yours,
Yan Jiang
College of Agriculture, Yangzhou University
Yangzhou 225009, Jiangsu, China